# Efficacy and Durability of Immune Response after Receipt of HPV Vaccines in People Living with HIV

**DOI:** 10.3390/vaccines11061067

**Published:** 2023-06-05

**Authors:** Cecilia Losada, Hady Samaha, Erin M. Scherer, Bahaa Kazzi, Lana Khalil, Ighovwerha Ofotokun, Nadine Rouphael

**Affiliations:** 1The Hope Clinic of the Emory Vaccine Center, Division of Infectious Diseases, Department of Medicine, Emory University, Decatur, GA 30030, USA; 2Division of Infectious Diseases, Department of Medicine, Emory University, Atlanta, GA 30322, USA

**Keywords:** human papillomavirus (HPV), HIV, HPV vaccine, immunogenicity, efficacy, vaccine durability

## Abstract

People living with HIV (PLH) experience higher rates of HPV infection as well as an increased risk of HPV-related disease, including malignancies. Although they are considered a high-priority group for HPV vaccination, there are limited data regarding the long-term immunogenicity and efficacy of HPV vaccines in this population. Seroconversion rates and geometric mean titers elicited by vaccination are lower in PLH compared to immunocompetent participants, especially in individuals with CD4 counts below 200 cells/mm^3^ and a detectable viral load. The significance of these differences is still unclear, as a correlate of protection has not been identified. Few studies have focused on demonstrating vaccine efficacy in PLH, with variable results depending on the age at vaccination and baseline seropositivity. Although waning humoral immunity for HPV seems to be more rapid in this population, there is evidence that suggests that seropositivity lasts at least 2–4 years following vaccination. Further research is needed to determine the differences between vaccine formulations and the impact of administrating additional doses on durability of immune protection.

## 1. Introduction

Human papillomavirus (HPV) is a common sexually transmitted infection that is associated with cervical cancer as well as other anogenital and oropharyngeal malignancies [1,2,3]. Three vaccines have been licensed in the United States (US): a bivalent vaccine targeting oncogenic types 16 and 18 (2vHPV), and a quadrivalent vaccine that provides additional protection against non-oncogenic types HPV-6 and HPV-11 (4vHPV), which were introduced more than 10 years ago [4,5]. A nonavalent vaccine that targets HPV types 6, 11, 16, 18, 31, 33, 45, 52 and 58 (9vHPV), has been the only HPV vaccine distributed in the United States since 2016 [1,6]. All these vaccines contain virus-like particles (VLPs) and are manufactured using recombinant DNA and eukaryotic cells [2,4,5,6]. Current Advisory Committee on Immunization Practices (ACIP) recommendations suggest a two or three-dose schedule depending on age and individual characteristics, including immune status at the time of vaccination [1].

These vaccines have proven to be effective in the general population, with a decrease in the prevalence of infections with HPV types included in the vaccine formulations as well as in the incidence of genital warts and HPV-associated cancers and pre-malignant conditions following vaccination campaigns [2,7,8,9,10,11,12]. Approval was granted based on clinical efficacy, to prevent infections as well as HPV-associated disease, as no serologic correlate of protection has been identified to date [2,13]. Current evidence suggests that vaccine protection is mainly mediated by neutralizing antibodies while cell-mediated immunity plays a more limited role [14].

Higher rates of HPV infection are reported in people living with HIV (PLH), as well as a more frequent carriage of multiple HPV types [2,15,16]. These findings are not only associated with increased exposure due to behavioral patterns but also with immune dysfunction and mucosal abnormalities [3,16]. A meta-analysis reviewing data on women living with HIV in low- and middle-income countries (LMICs) reported an HPV prevalence of 51% when considering only high-risk types [17]. Men who have sex with men (MSM) and living with HIV have the highest prevalence of HPV-16 infection and of anal HPV, regardless of the type [18,19,20]. 

PLH are also at increased risk of suffering HPV-related disease, including malignancies. Current evidence suggests the more rapid progression of precancerous lesions and higher rates of HPV persistence in this population [2,21,22,23]. Regarding the risk of cervical cancer, an acquired immunodeficiency syndrome (AIDS)-defining condition, data show an overall 6-fold higher risk of developing this malignancy and a shorter interval between infection and the development of invasive carcinoma, with an average age of diagnosis approximately 5–10 years younger than women not living with HIV. Moreover, the magnitude of decline in cervical cancer incidence after the introduction of triple antiretroviral therapy (ART) has not been as high as the reduction evidenced for other AIDS-defining malignancies, especially in LMICs [3]. Similarly, anal carcinoma is the fourth most common type of cancer in PLH, with an estimated incidence of 89 per 100,000 in MSM living with HIV compared to 1.6 per 100,000 person-years (PY) in the general population [24].

As a result of the increased prevalence of HPV infection and disease discussed above, PLH are considered a high-priority group for HPV vaccination [2,25]. However, only studies with small sample size have contributed to current knowledge regarding the safety and immunogenicity of HPV vaccines in this population [26,27,28,29,30,31]. More information is still needed to draw conclusions about optimal vaccine type and dosing as well as clinical efficacy in this population. 

HIV infection is associated with altered cellular and humoral responses to vaccines, even in individuals on ART with an undetectable viral load (VL) and CD4 counts that exceed 500 cells/mm^3^ [16]. The durability of immune response to vaccination in PLH has been addressed in several studies, with evidence of more rapidly waning immunity compared to uninfected participants [32]. Recent studies have evaluated strategies to optimize immune response to vaccines in PLH, such as changes in doses and schedules, the use of adjuvants and identifying the optimal timing of vaccination considering the patient’s immune status [13,16,32].

In this review, we aim to provide a summary of current knowledge regarding HPV vaccine efficacy (VE) and immunogenicity in PLH, focusing on immune response durability and possible strategies to improve it.

## 2. Guidelines for HPV Vaccination in PLH

All guidelines emphasize the importance of HPV vaccination in PLH and agree on maintaining the three-dose schedule in this population (Table 1).

Centers for Disease Control and Prevention (CDC) guidelines recommend giving a series of three intramuscular HPV vaccine injections to all PLH (men and women) up to the age of 26 that have not already been vaccinated at the target age for immunization (11–12 years old). Additionally, the guidelines suggest that some adults aged 27–45 might still benefit from vaccination and are eligible to receive the vaccine after discussion with their medical provider [1,15]. 

The updated European AIDS Clinical Society (EACS) guidelines published in 2022 recommend the administration of three doses of HPV vaccine between the ages of 9 and 45, using the nonavalent vaccine if it is available [33]. Similarly, the British HIV association (BHIVA) suggests in its 2015 recommendations that previously unvaccinated PLH aged up to 26 years should be offered HPV vaccination, extending the age to 40 years old in the case of women and MSM [25]. This guideline recommends vaccination regardless of the CD4 cell count, ART use and VL, except for ART-naïve individuals with CD4 counts below 200 cells/mm^3^ who might benefit from deferring vaccination until they are established on ART.

The updated World Health Organization (WHO) recommendations, published in December 2022, include an alternative single-dose schedule as an off-label option that can be used in immunocompetent girls and boys aged 9–20 years [2]. They suggest that this regimen has comparable efficacy and durability as the two-dose schedule and may have operational and financial advantages, possibly resulting in increased coverage in LMICs [2,34,35,36]. However, studies have shown lower antibody titers as well as reduced B and T-cell responses in one-dose recipients [34,37,38,39,40] and concerns remain regarding their potential impact on waning immunity [41]. This simplified schedule is not suitable for immunocompromised individuals, such as PLH, who are recommended to receive three doses when possible, or at least two doses separated by 6 months [2].

## 3. Safety of HPV Vaccines in HIV-Infected Adults

Publications regarding HPV vaccination in PLH reported that the bivalent, quadrivalent and nonavalent vaccines were safe and tolerable for this population compared to the results in studies including immunocompetent participants [2,42,43]. This was concluded after reviewing safety data from the use of these vaccines in females and males as well as children in clinical trials and post-licensure surveillance. The number of serious adverse events (SAEs) reported in studies including PLH are similar regardless of the vaccine type received, with incident rates of SAEs mostly ranging from 0 to 7% [13]. Moreover, evidence suggests that vaccination does not affect HIV disease progression, as no significant variations in CD4 counts and VL were seen across multiple studies [26,28,42].

## 4. Immunogenicity of HPV Vaccines in HIV-Infected Adults

HPV vaccines result in a robust and durable neutralizing antibody response, with higher titers than natural infection. Peak antibody titers reported in the clinical trials of mostly immunocompetent individuals are observed 4 weeks after first vaccination. Seropositivity rates and geometric mean titers (GMTs) remain high for at least 12 years following the receipt of a multidose schedule of 2vHPV and 4vHPV and for 7 years after the most recently approved nonavalent formulation [2,41,44,45,46,47,48]. 

Both 2vHPV and 4vHPV demonstrated cross-protection against HPV types not included in the vaccine [2]. However, some studies have shown that this protection is highly variable and less durable than immune responses toward HPV types included in the vaccines [49].

In studies evaluating these vaccines in PLH, almost all HPV seronegative participants seroconverted to HPV types included in the vaccines and were still seropositive 28 weeks after initial vaccination. Table 2 summarizes the results from randomized controlled trials (RCTs) assessing immunogenicity in PLH. A recent meta-analysis reported a pooled proportion of seropositivity by 28 weeks following three doses of bivalent, quadrivalent and nonavalent formulations of 0.99 (95% CI, 0.95–1.00), 0.99 (95% CI, 0.98–1.00) and 1.00 (95% CI, 0.99–1.00) for HPV-16, and 0.99 (95% CI, 0.96–1.00), 0.94 (95% CI, 0.91–0.96) and 1.00 (95% CI, 0.99–1.00) for HPV-18, respectively [13].

Data from two meta-analyses also show that GMTs are lower for HPV-18 compared to HPV-16, both initially and during follow-up [13,42]. The use of 2vHPV in PLH results in similar seroconversion rates for HPV-16 compared to 4vHPV but is associated with higher seroconversion rates and average GMTs for HPV-18 [13,29,50]. A study including 91 PLH, who were randomized to receive 4vHPV or 2vHPV, showed that the latter induced seroconversion in 100% of participants, while in the quadrivalent vaccine-arm, 96% seroconverted to HPV-16 but only 73% converted to HPV-18 [29]. Published data from the same study group revealed cross-protection against HPV-31, HPV-33 and HPV-45 for both vaccine groups [51].

When comparing immunogenicity in PLH and immunocompetent participants, current data show slightly lower seroconversion rates and GMT titers in PLH [16]. Differences in seropositivity reported in a recent meta-analysis were statistically significant for HPV-16, HPV-18 (28 weeks, 29–99 weeks) and HPV-6 (29–99 weeks) between the two groups, with the largest difference reported at 29–99 weeks for HPV-18 [13]. 

A prospective cohort of perinatally HIV-infected youth showed seroconversion rates of 83%, 84%, 90% and 62% for HPV-6, HPV-11, HPV-16 and HPV-18 after the receipt of quadrivalent vaccine, which were significantly lower than the rates in perinatally exposed but uninfected participants [52]. GMTs were also lower for each of the HPV types and following every dose. A randomized controlled trial (RCT) in South Africa including 120 women living with HIV (WLH) aged 18–25 years old who were randomized to three doses of 2vHPV vs. placebo and 30 vaccinated uninfected controls, showed lower antibody titers in WLH, although all vaccinated women remained seropositive after 12 months [28]. These results might have been influenced by high levels of HPV seropositivity at baseline. Similarly, a study evaluating 4vHPV safety and immunogenicity in well-controlled PLH aged 7–12-years-old revealed that seroconversion to all four antigens occurred in more than 96% of vaccine recipients [26]. However, GMTs against HPV-6 and HPV-18 were 30–50% lower compared to those reported in historical cohorts for that age group.

The significance of these lower antibody titers is yet to be determined, as a correlate of protection has not been identified, whilst antibody levels needed to achieve protection against HPV acquisition and disease are still unknown [13,41,53,54]. The contribution of cellular immune responses to HPV protection in this population is also unclear. A study on well-controlled PLH revealed that 4vHPV and 2vHPV induced comparable T-cell responses at 12 months after vaccination and detected no clear correlation with antibody titers [55].

**Table 2 vaccines-11-01067-t002:** Summary of evidence from randomized controlled trials assessing the immunogenicity of HPV vaccines in PLH.

Study	Population	N	Intervention	Median CD4 Count (Cells/mm^3^)	% On ART	Follow-Up (after Dose 1)	Seropositivity in Vaccinated Group at the End of Follow-Up	Other Important Findings
Toft et al. (2014) [50] NCT01386164	Adults (median age 2vHPV group: 47 years; median age of 4vHPV group: 44.5 years)	91	Randomized 1:1–3 doses of 2vHPV or 4vHPV	2vHPV group: 600 4vHPV group: 585	2vHPV group: 88.9% 4vHPV group: 87%	12 months	Results reported in Faust et al. [29]	No significant differences in HPV-16 antibody titers were detected, but HPV-18 titers were higher in the 2vHPV group, with a GMT ratio at 12 months of 4.15 (95% CI, 1.95–8.84). Women had higher antibody titers than men after receipt of 2vHPV.
Faust et al. (2016) [29] NCT01386164	Adults (mean age 46 years)	91	Randomized 1:1–3 doses of 2vHPV or 4vHPV	See Toft et al. [50]	See Toft et al. [50]	12 months	2vHPV: 100% seroconverted for HPV-16 and HPV-18 4vHPV: 96% seroconverted for HPV-16 and 73% for HPV-18	Cross reactive antibodies were detected after vaccination. The differences in seroconversion rate between vaccines were statistically significant for HPV-6 and 11 (higher for 4vHPV), as well as HPV-18 and 45 (higher for 2vHPV)
Zurek Munk-Madsen et al. (2018) [55] NCT01386164	Adults (median age 2vHPV group: 46 years; median age of 4vHPV group: 45 years)	30	2vHPV or 4vHPV with cellular immunogenicity analysis in a subset of participants from NCT01386164	2vHPV group: 610 4vHPV group: 620	100%	12 months	Not reported	T-cell frequencies increase significantly after vaccination with 2vHPV or 4vHPV. Comparable T-cell responses induced by the 2 vaccines at 12 months. No clear correlation between T-cell responses and antibody titers.
Denny et al. (2013) [28] NCT00586339	Women aged 18–25 years (mean age 22 years)	120 WLH Comparator group: 30 HIV-negative women	Randomized 1:1–3 doses of 2vHPV or placebo. Comparator group: 2vHPV	2vHPV group: 451 Placebo group: 458	1.7%	12 months	HPV-16: 100% HPV-18: 100%	High % of seropositivity at baseline. HPV-16 and HPV-18 antibody titers remained above levels associated with natural infection at 12 months. Vaccination resulted in sustained HPV-16 and HPV-18 CD4 T-cell responses. No impact of baseline CD4 count or HIV VL on the magnitude of the immune response.
Levin et al. (2010) [26] IMPAACT P1047 Study NCT00339040	Children aged 7–12 years	126	Randomized 3:1– 3 doses of 4vHPV or placebo	4vHPV group: 868 Placebo group: 1013	Not reported	28 weeks	HPV-6: 100% HPV-11: 100% HPV-16: 100% HPV-18: 97%	GMTs against HPV-6 and HPV-18 were 30–50% lower compared to those reported in historical healthy cohorts for that age group. Univariate linear regression analysis showed significant correlations between higher antibody levels and lower VL (<5000 vs. >5000 copies/mL) as well as lower CD8%. No consistent association with CD4% was found.
Folschweiller et al. (2020) [56] NCT01031069	Women aged 15–25 years (mean age 19.8 years)	257 WLH Comparator group: 289 HIV-negative women	Randomized 1:1 to receive 2vHPV or 4vHPV	586.5	61.5%	24 months	Seroconversion rates 2vHPV group (WLH): 98.8% for HPV-16 and 95.2% for HPV-18 Seroconversion rates for 4vHPV group (WLH): 96.4% for HPV-16 and 68.4% for HPV-18	GMTs against HPV-16 and HPV-18 were 2.74- and 7.44-fold higher, respectively, after the receipt of 2vHPV vs. 4vHPV (*p* < 0.0001) at 7 months (PBNA) in WLH. Similar results up to 24 months reported by ELISA. Antibody responses were overall lower in WLH vs. comparator group. Higher antibody responses were detected in WLH with higher CD4 counts. Trend for higher memory B-cell responses with 2vHPV vs. 4vHPV.

HPV, human papillomavirus; PLH, people living with HIV; N, number of participants (vaccinated); ART, antiretroviral therapy; 2vHPV, bivalent HPV vaccine; 4vHPV, quadrivalent HPV vaccine; GMT, geometric mean titers; WLH, women living with HIV; VL, viral load; PBNA, pseudovirion-based neutralization assay; ELISA, enzyme-linked immunosorbent assay.

A publication looking at sera from 75 PLH and 147 immunocompetent men who received 3 doses of quadrivalent vaccine reported that antibody avidity was significantly lower for HPV-16 and HPV-18 one month after the third vaccine dose in PLH [57]. However, the avidity levels remained high for both groups and absolute differences were small. The clinical importance of antibody avidity in HPV VE is yet to be determined.

Most studies conclude that PLH with CD4 counts below 200 cells/mm^3^ and detectable VL have lower seropositivity rates and GMTs compared to the well-controlled PLH, as seen in other vaccine platforms [31,57,58]. However, a study in young WLH in South Africa found no impact of the baseline CD4 count or VL on the magnitude of antibody response against HPV-16 and HPV-18 following 2vHPV vaccination [28]. In another publication regarding immune response in HIV-infected children following quadrivalent vaccine, the univariate analysis showed significant correlations between the higher antibody levels and lower VL (<5000 vs. >5000 copies/mL) as well as lower CD8%, but no consistent association with CD4% was found [26]. Further research is needed to understand the influence of CD4 counts, VL and ART on the vaccine response, especially on participants reconstituting the immune status despite HIV infection.

## 5. Durability of Immune Protection following HPV Vaccination in PLH

The long-term durability of the immune response in PLH has not been sufficiently studied. Although seropositivity rates and antibody titers decline over time, there is evidence that suggests that seropositivity lasts at least 2–4 years [2,13]. Table 3 summarizes evidence of the studies reporting the durability of HPV vaccine immune response after more than 2 years since initial vaccination in PLH.

**Table 3 vaccines-11-01067-t003:** Summary of evidence from studies reporting the durability of the HPV vaccine response after more than two years since initial vaccination in PLH.

	Population	N	Vaccine	Study Design	Median CD4 Count (Cells/mm^3^)	% on ART	Follow-Up	Seropositivity at the End of Follow-Up	Other Important Findings
Mugo et al. (2021) [30]	CLH aged 9–14 at time of vaccination	176	4vHPV (3 doses)	Re-enrollment 2 years after vaccination for extended follow-up	701 (at re-enrollment)	94.9% (at re-enrollment)	4 years after initial vaccination	HPV-6: 83% HPV-11: 80% HPV-16: 90% HPV-18: 77%	Plateau in mean log10 antibody titers at month 24. Children with detectable VL were less likely to sustain seropositivity at month 48.
Levin et al. (2017) [53] IMPAACT P1085 Study NCT01206556	CLH previously enrolled in NCT00339040 (median age: 12 years)	97 (74 in 4-dose arm and 23 in 3-dose arm)	4vHPV (total of 3 or 4 doses)	Arm A: 4th dose at week 96 Arm B: previous placebo recipients began 3-dose schedule	4-dose group: 710 3-dose group: 749	4-dose group: 99% 3-dose group: 91%	4–5 years after last vaccination	4-dose group (LIA): HPV-6: 95% HPV-11: 97% HPV-16: 100% HPV-18: 88% 3-dose group (LIA): HPV-6: 86% HPV-11: 71% HPV-16: 100% HPV-18: 71%	GMTs for HPV-16 and HPV-18 were higher in the 4-dose group, but no significant differences in seropositivity rates were detected and the rate of decline was similar in both groups. Antibody titers correlated strongly with low VL, low CD8% and high CD4%.
Weinberg et al. (2018) [59] IMPAACT P1085 Study NCT01206556	CLH. Median age: 12 years	97 (74 in 4-dose arm and 23 in 3-dose arm)	4vHPV (total of 3 or 4 doses)	Cellular immunogenicity analysis in a subset of participants from IMPAACT P1085	4-dose group: 710 3-dose group: 749	4-dose group: 99% 3-dose group: 91%	4–5 years after last vaccination	Results reported in Levin et al. (2017) [53]	Decrease in global T-cell function and HPV-specific memory B-cells over time. No significant differences in B and T-cell memory between 3 and 4 doses. HPV-16-specific IL2 and IFNγ T-cell responses were significantly higher than those for HPV-18 at most time points. Decline in HPV-specific memory B-cells reached statistical significance for HPV-16 in the 4-dose group. Correlation between T-cell response and antibody titers (cLIA) for HPV-16 and HPV-18 in both arms, but not with memory B-cells. Positive correlation of T-cell response with CD4 counts and percentages, and negative correlation with plasma HIV VL and with CD8 counts and percentages. No consistent association between memory B-cell response and degree of HIV control.
Moscicki et al. (2019) [52]	Perinatally infected (PHIV) or perinatally exposed (PHEU) youth (mean age of 1st vaccine dose: 13.3 years)	310 PHIV 148 PHEU	4vHPV (32% received at least two doses and 14% unvaccinated)	Prospective observational cohort	Not reported	Not reported	Mean of 2.9 years since last vaccine dose	Results in PHIV cohort (vaccinated only): HPV-6: 83% HPV-11: 84% HPV-16: 90% HPV-18: 62%	PHIV had significantly lower seropositivity rates than PHEU for each HPV-type. PHIV had lower GMTs regardless of dose. Unvaccinated participants in both cohorts had a markedly lower % of seropositivity than vaccinated ones. Higher antibody titers associated with younger age, lower VL and higher CD4% at the time of first vaccination as well as shorter interval between the last dose and specimen collection. Abnormal cytology detected in 33 of 56 PHIV and 1 of 7 PHEU sexually active vaccinated female participants.

HPV, human papillomavirus; PLH, people living with HIV; N, number of participants; ART, antiretroviral therapy; CLH, children living with HIV; 4vHPV, quadrivalent HPV vaccine; VL, viral load; LIA, line-immunoassay; GMT, geometric mean titers; IL-2, interleukin-2; IFNγ, interferon gamma; cLIA, competitive Luminex immunoassay; PHIV, perinatally HIV-infected; PHEU, perinatally HIV-exposed uninfected.

A recent publication with data on the 4-year follow-up of 176 children living with HIV (CLH) aged 9–14 in Kenya after the receipt of the quadrivalent vaccine reported that the rate of seropositivity at 48 months was 83% for HPV-6, 80% for HPV-11; 90% for HPV-16; and 77% for HPV-18 [30]. A plateau in mean log10 antibody titers was detected between months 24 and 48, which remained higher in participants with an undetectable viral load.

Likewise, a 4–5 year follow-up of CLH after the receipt of 3 doses of 4vHPV in another study reported that seropositivity levels measured with a line-immunoassay (LIA) were 86%, 71%, 100% and 71% for HPV types 6, 11, 16 and 18, respectively, experiencing a slightly higher reduction than in the one reported in similar studies in healthy children [53]. GMTs also declined in CLH between 2 and 4–5 years post-vaccination, similarly to the decline evidenced in immunocompetent children.

A phase 2 open-label study evaluating the response to the quadrivalent HPV vaccine in young WLH evidenced that GMTs declined significantly for all four vaccine-types, but remained higher than those induced by natural infection, consistent with the studies in healthy women [27].

Several studies reporting results based on a competitive luminex immunoassay (cLIA) suggest that waning humoral immunity might be more rapid for HPV-18 compared to other types [13,27,53]. However, these results should be interpreted with caution, as the assay is known to underrepresent the total HPV-18 antibodies [14,41,60,61,62]. This finding is supported by the fact that protection against HPV-18 remains robust despite a pronounced drop in antibody titers over time reported in studies using cLIA including immunocompetent participants [63,64]. 

Einstein et al. reported the differences in antibody titers between 2vHPV and 4vHPV in immunocompetent women aged 18–45 which were seronegative and DNA-negative at baseline [65,66]. At 60-month follow-up, GMTs were 2.3–7.8-fold (HPV-16) and 7.8–13-fold (HPV-18) higher across different age groups in recipients of 2vHPV compared to individuals vaccinated with 4vHPV. Statistical modeling performed in the study predicted that the immune response lasts longer after vaccination with the bivalent formulation. Furthermore, studies have shown that these vaccines induce memory B-cell responses of greater magnitude for HPV-16 than HPV-18 but evidenced conflicting results when comparing 2vHPV and 4vHPV [66,67,68].

There are limited data comparing different vaccine formulations regarding long-term immunogenicity in PLH. An RCT reporting results from 91 PLH found no significant differences in HPV-16 antibody titers, but HPV-18 titers were higher in the 2vHPV compared to the 4vHPV group, with a GMT ratio at 12 months of 4.15 (95% CI, 1.95–8.84) [50]. In this study, women had higher antibody titers than men after the receipt of 2vHPV, but no sex-specific differences were detected following the administration of 4vHPV. Another study reported higher GMTs as well as a trend towards higher memory B-cell responses in WLH at 2-year follow-up after the receipt of the bivalent formulation, with greater differences noticed for HPV-18 [56]. Long-term follow-up is needed to compare antibody decline and memory B-cell responses for vaccine formulations over time in PLH and to determine its potential impact on VE.

Regarding cellular immunity, a study including CLH who received three or four doses of quadrivalent HPV vaccine reported no significant differences between treatment groups and demonstrated that B- and T-cell responses can be detected 5 years after vaccination [59]. HPV-specific memory B-cells tended to decrease between 2 and 5 years after the last vaccine dose and reached statistical significance for HPV-16 in the four-dose group. T-cell function declined over time and T-cell responses were higher for HPV-16 compared to HPV-18. Studies with a longer follow-up and including participants recently starting ART after diagnosis are needed to further characterize the cellular immune responses to HPV vaccine in PLH.

More information regarding long-term immunogenicity in PLH is required to determine vaccine durability and its association with VE, as there are no data on seropositivity rates and GMTs beyond 4–5 years following vaccination. Antigen re-exposure by the administration of additional doses triggers an anamnestic response in vaccinated individuals, as shown in studies including PLH as well as in immunocompetent individuals [59,69,70,71]. However, the ability of natural infection to elicit similar memory B-cell responses and its contribution to long-term protection is still unclear [14,27,67]. 

As a result of the wide availability of ART, the life expectancy of PLH has significantly increased in developed countries, evidenced by the fact that more than half of PLH in the US are over 50 years old [72]. The combined impact of age-associated immunosenescence with HIV infection on the HPV vaccine response and the long-term durability of immune protection are yet to be determined. Since HPV vaccines were introduced in 2006 and are mainly indicated in teenagers and young adults, there are no data regarding the persistence of the immune response in older PLH.

## 6. Clinical Efficacy of Quadrivalent HPV Vaccines in PLH

There is evidence that VE against HPV vaccine-type infection as well as cervical, vaginal and vulvar neoplasia in women and anogenital neoplasia in men, remains high 10 years after vaccination in immunocompetent individuals [2,73]. However, only a few studies focused on demonstrating VE in PLH [52,74,75,76]. Earlier studies showed modest VE in preventing HPV infection and disease, probably related to vaccine administration after HPV exposure, demonstrated by high baseline seropositivity, as well as the late introduction of ART [16].

McClymont et al. reported results from a Canadian cohort of WLH aged 9–65 years old [74]. A total of 279 WLH were vaccinated with the quadrivalent HPV vaccine and followed for a median of 2 years. Among them, 95.3% received 3 vaccine doses. The median CD4 count was 500 cells/mm^3^ and 69% of the participants had undetectable VL. The incidence rate (IR) of newly acquired persistent vaccine-type HPV was 2.3 per 100 PY (95% CI, 1.1–4.1) in the intention to treat (ITT) group, more than half of which are due to HPV-18, and 1.0 per 100 PY (95% CI, 0.3–2.6) in the per-protocol efficacy (PPE) population. No cases of vaccine-type HPV-associated cervical intraepithelial neoplasia (CIN) of grade two or higher developed in women with normal baseline cytology and the IR for anogenital warts was 2.3 per 100 PY (95%CI, 1.2–4.1) in the ITT group. When compared with reported rates of infection and disease in other studies, WLH seem to be at higher risk for vaccine failure, but rates of persistent HPV infection are lower than in unvaccinated WLH, suggesting an important benefit from vaccination. 

A prospective cohort study comparing the immunogenicity and VE of the quadrivalent vaccine in perinatally HIV-infected and exposed-uninfected youth revealed an IR of 15.0 (95% CI, 10.9–20.6) and 2.9 (95% CI, 0.4–22.3) per 100 PY of abnormal cervical cytology in sexually active HIV-positive and -negative participants, respectively [52]. 

An RCT in PLH aged 27 or older analyzing VE for the prevention of persistent or newly acquired anal infections as well as high-grade squamous intraepithelial lesions (HSIL) had to be stopped early due to futility [75]. A total of 575 participants were randomized (1:1) to receive quadrivalent vaccine or placebo. VE was 22% (95.1% CI, −31%–53%) for the prevention of persistent anal infection or single detection at the final visit and 0% (95% CI, −44%–31%) for improving HSIL outcomes. High rates of baseline HPV infection were reported in the study, highlighting the importance of vaccination before the start of sexual activity and HPV exposure. Similarly, another RCT carried out in Spain included 129 MSM PLH of more than 26 years of age who were randomized to receive the quadrivalent vaccine or placebo [76]. Both arms did not significantly differ in rates of ≥HSIL (14.1 vs. 13.1%, *p* = 0.98), external anogenital lesions (11.1 vs. 6.8%, *p* = 0.4) or in the acquisition of vaccine-type HPV after 48 months. However, the vaccine offered protection against the acquisition of genotype 6 during the first year of follow-up (7.5 vs. 23.4%, *p* = 0.047).

## 7. Possible Strategies to Optimize Immune Response in PLH

Several strategies to optimize HPV vaccine response and improve the durability of protection have been analyzed in previous publications (Table 4).

Superior immunogenicity has been reported with the use of 2vHPV, a Toll-like receptor (TLR)-adjuvanted vaccine, compared to 4vHPV, an alum-adjuvanted vaccine, especially when considering memory B-cell responses to HPV-18 [16,56]. 2vHPV contains AS04, which consists of monophosphoryl lipid A (MPL), a derivative of the lipopolysaccharide molecule of *Salmonella minnesota*’s bacterial wall and adsorbed onto aluminum hydroxide [4,78]. This adjuvant stimulates TLR in antigen presenting cells, inducing their migration to lymph nodes and the release of cytokines that stimulate antibody and cellular immune responses [28,78]. This strategy is consistent with data reported from other vaccine platforms analyzing immune responses in PLH, such as Hepatitis B [79,80,81]. Adjuvants can contribute to by-pass defects in the memory T-cell follicular function seen in PLH and might potentially contribute to obtaining improved immunogenicity and more durable responses. However, no significant differences in VE have been demonstrated across different vaccine platforms. In addition, 2vHPV does not provide direct protection against HPV-6 and HPV-11, types commonly associated with genital warts and respiratory papillomatosis. A mixed vaccine schedule alternating 2vHPV and 4vHPV/9vHPV could overcome this difficulty [16].

9vHPV has a higher antigen content than the quadrivalent vaccine and its administration resulted in slightly higher GMTs in the general population for HPV-18 [77]. It remains to be determined whether this is also evidenced in PLH and if it can contribute to improving the durability of immune responses. Clinical trials analyzing the safety, immunogenicity and VE of the nonavalent vaccine in this population are currently underway and their results will help guide future recommendations.

The impact of administrating a fourth dose of the quadrivalent vaccine at 72 weeks after last vaccination was studied in CLH [53]. This intervention led to significantly higher antibody titers at the 4–5-year follow-up, but no difference in seropositivity was detected and it did not improve the rate of antibody decline over time. The administration of an additional dose was safe and resulted in an anamnestic response at 7 days after vaccination. To our knowledge, higher-dose formulations and alternative routes of administration have not been studied as possible strategies to enhance immune responses to vaccination.

Another strategy to improve the durability of protection might be delaying vaccination until immune reconstitution has been achieved, as most studies report enhanced immunogenicity after ART initiation, with optimized CD4 counts and undetectable VL [27,31,53,57,58].

## 8. Conclusions

Although PLH are considered a high-priority group for HPV vaccination, there are only modest data regarding long-term immunogenicity and efficacy in this population. Studies that address the durability of immune response and its potential impact on VE should be considered research priorities, as there is currently insufficient information on both cellular and humoral responses beyond 2 years after vaccination. Further data are necessary to understand the influence of CD4/CD8 counts and HIV VL on vaccine response, especially in individuals who have experienced immune reconstitution as a result of ART. Studies focusing on HIV-positive transgender individuals as well as persons who inject drugs are also needed to understand the impact of vaccination in these populations. RCTs comparing VEs between different vaccine formulations and schedules in PLH are essential to dictate future recommendations. Optimized vaccination strategies, in combination with improved screening and treatment procedures, can potentially reduce the burden of HPV disease in PLH.

## Figures and Tables

**Table 1 vaccines-11-01067-t001:** HPV vaccine recommendations for PLH.

	CDC/ACIP [1,15]	EACS 2022 [33]	BHIVA 2015 [25]	WHO [2]
Recommended age for administration	11 or 12 (9–26)	According to national guidelines (9–45)	12–13	9–14
Age limit for catch-up vaccination	26 (persons aged 27–45 might be eligible after shared clinical decision making)	45	26 (extended to 40 years of age for women and MSM)	Not specified
Number of doses	3	3	3	3 (preferred) or an alternative 2-dose regimen with minimum 6-month interval
Preferred HPV vaccine	9vHPV (only vaccine currently distributed in US)	9vHPV	9vHPV	2vHPV, 4vHPV or 9vHPV
CD4 count and VL at time of vaccination	No specific recommendation	Preferably after achieving suppressed VL and immune reconstitution (CD4 count > 200 cells/mm^3^)	In ART-naïve patients with CD4 cell counts below 200 cells/mm^3^, vaccination may be deferred until the patient is established on ART	No specific recommendation

HPV, human papillomavirus; PLH, people living with HIV; CDC, Centers for Disease Control and Prevention; ACIP, Advisory Committee on Immunization Practices; EACS, European AIDS Clinical Society; BHIVA, British HIV association; WHO, World Health Organization; MSM, Men who have sex with men; 2vHPV, bivalent HPV vaccine; 4vHPV, quadrivalent HPV vaccine; 9vHPV, nonavalent HPV vaccine; US, United States; VL, viral load; ART, antiretroviral therapy.

**Table 4 vaccines-11-01067-t004:** Possible strategies to optimize HPV vaccine response in PLH.

Strategy	Benefits	Limitations	Feasibility
Use of 2vHPV (adjuvanted) vs. 4vHPV/9vHPV	AS04 adjuvant in 2vHPV stimulates TLR in antigen presenting cells, leading to enhanced humoral and cellular responses Studies in immunocompetent participants and PLH have evidenced higher GMTs, especially against HPV-18 after vaccination with 2vHPV [29,50,65,66] Possibility of cross-protection against HPV-types not included in the vaccine [29]	No significant differences in VE have been evidenced across vaccine platforms for immunocompetent participants [2] To our knowledge, there are no RCTs comparing VE of the two formulations in PLH No direct protection against HPV-6 and HPV-11, types associated with genital warts and respiratory papillomatosis No RCTs up to date comparing the immunogenicity of 9vHPV vs. 2vHPV in PLH	No major obstacles to implementation Possibility of using a mixed-dose schedule
Use of 9vHPV vs. 2vHPV/4vHPV	9vHPV has a higher antigen content than 4vHPV and has led to slightly higher GMTs for HPV-18 in immunocompetent participants [77] Direct protection against a higher number of HPV types	No significant differences in VE or durability (for HPV-16 and HPV-18) have been evidenced across vaccine platforms for immunocompetent participants [2] Studies assessing 9vHPV in PLH are currently underway, limiting information on immunogenicity and VE	9vHPV is currently the only vaccine distributed in many countries Higher cost per vaccine may hinder implementation in LMICs with higher HIV burden
Additional vaccine dose (4-dose regimens)	Study in CLH has shown an anamnestic response 7 days after vaccination with 4th dose (at week 96) [53]	Same study showed no significant differences in seropositivity or antibody decline rates for HPV-16 and HPV-18 [53] There are no RCTs comparing the VE of 3- vs. 4-dose schedules in PLH Studies in immunocompetent participants have shown similar VEs to regimens with reduced number of doses [34,35,36]	Higher cost of additional dose may hinder implementation in LMICs, with a higher HIV burden
Delaying vaccination until immune reconstitution	PLH with CD4 counts below 200 cells/mm^3^ and detectable VL have lower seropositivity rates and GMTs compared to well-controlled PLH [31,57,58]	Conflicting results regarding influence of CD4/CD8 counts as well as HIV VL on vaccine response in subjects with a reconstituted immune status despite HIV infection [26,28] Missed opportunity of vaccination	EACS 2022 [33] and BHIVA 2015 [25] guidelines suggest waiting until immune reconstitution

HPV, human papillomavirus; PLH, people living with HIV; 2vHPV, bivalent HPV vaccine; 4vHPV, quadrivalent HPV vaccine; 9vHPV, nonavalent HPV vaccine; TLR, Toll-like receptor; GMT, geometric mean titers; VE, vaccine efficacy; RCTs, randomized controlled trials; LMICs, low- and middle-income countries; CLH, children living with HIV; VL, viral load; EACS, European AIDS Clinical Society; BHIVA, British HIV association.

## Data Availability

Not applicable.

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
