# Peer review of "Efficacy and Durability of Immune Response after Receipt of HPV Vaccines in People Living with HIV"

_vaccines, 2023, doi:10.3390/vaccines11061067_

Round 1

Reviewer 1 Report

This is a useful review of published data on immune responses to HPV vaccines in people living with HIV. The topic is an important one as it has clinical relevance and there are significant gaps in knowledge about the utility of these vaccines in PLH. This review is comprehensive but could benefit from better organization. It mixes studies in adults and children, does not systematically identify impact of immune status on vaccine response. It appears most studies are immunogenicity only but hard to sort out immunogenicity and efficacy data. It does review alternative strategies which appear to impact durability of immune responses but it is not clear which if any alter efficacy. A table summarizing the data from the various trials may help improve readability, and a more detailed list of gaps to be addressed in future studies in the conclusions would be useful. 

There are a few minor grammatical and spelling errors

line 107 "wanning" instead of waning

line 113: remove "as"

line 130: immune response(s )(should be pleural)

line 146: remove "also"

Author Response

Comments and Suggestions for Authors: This is a useful review of published data on immune responses to HPV vaccines in people living with HIV. The topic is an important one as it has clinical relevance and there are significant gaps in knowledge about the utility of these vaccines in PLH. This review is comprehensive but could benefit from better organization. It mixes studies in adults and children, does not systematically identify impact of immune status on vaccine response. It appears most studies are immunogenicity only but hard to sort out immunogenicity and efficacy data. It does review alternative strategies which appear to impact durability of immune responses but it is not clear which if any alter efficacy. A table summarizing the data from the various trials may help improve readability, and a more detailed list of gaps to be addressed in future studies in the conclusions would be useful. 

Response: Thank you for reviewing the manuscript and providing comments. We have included 2 tables to summarize data from available studies to improve readability. Table 2 provides a summary of evidence from Randomized Clinical Trials assessing immunogenicity of HPV vaccines in PLH while Table 3 outlines studies reporting durability of HPV vaccine immune response beyond two years from initial vaccination. Additionally, both tables provide information on available data from the studies described regarding the impact of immune status on vaccine response. We have also added Table 4, which describes possible strategies to improve vaccine response in this population. We have provided a more detailed list of gaps in current knowledge to be addressed in future studies in the conclusion, as suggested.

Comments on the Quality of English Language: There are a few minor grammatical and spelling errors

line 107 "wanning" instead of waning

line 113: remove "as"

line 130: immune response(s )(should be pleural)

line 146: remove "also"

Response: Thank you for pointing out these mistakes. We have corrected spelling and grammatical errors mentioned above, except for the spelling of “waning” which we believe is correct.

Reviewer 2 Report

1.     Please provide more evidences to support the decreasing of immunogenicity of HPV vaccines is related to CD4+ T cells or CD8+ T cell or both.

2.     Please provide a Table to compare different types of HPV vaccines (e.g., nonvalent formulation, 2vHPV, 4vHPV and 9vHPV) for the immunogenicity of HPV vaccines in HIV-infected adults.

3.     Please provide a Table to compare the HPV vaccines produced by different manufacturers to verified the durability of immune protection following HPV vaccination in PLH.

4.     Please make a Table to summarize and compare the possible strategies to optimize immune response of HPV vaccines in PLH, including feasibility, prospects, limitation and current development.

No

Author Response

Point 1: Please provide more evidences to support the decreasing of immunogenicity of HPV vaccines is related to CD4+ T cells or CD8+ T cell or both.

Response: Thank you for the comment. We have included additional information regarding the impact of CD4/CD8 counts on vaccine response in Tables 2 and 3.

Point 2: Please provide a Table to compare different types of HPV vaccines (e.g., nonvalent formulation, 2vHPV, 4vHPV and 9vHPV) for the immunogenicity of HPV vaccines in HIV-infected adults.

Response: Thank you for the suggestion. We have added Table 2 that provides a summary of evidence from Randomized Controlled Trials (RCTs) assessing immunogenicity of HPV vaccines (2vHPV, 4vHPV) in PLH. To our knowledge, there are no published RCTs reporting on immune response to 9vHPV in PLH.

Point 3: Please provide a Table to compare the HPV vaccines produced by different manufacturers to verified the durability of immune protection following HPV vaccination in PLH.

Response: Thank you for the comment. We have added Table 3 that outlines studies reporting durability of HPV vaccine immune response beyond two years from initial vaccination in PLH. Unfortunately, these results are limited to studies reporting on immune response to 4vHPV, as to our knowledge there is no published data regarding immunogenicity results in PLH beyond two years from vaccination for the other formulations.

Point 4: Please make a Table to summarize and compare the possible strategies to optimize immune response of HPV vaccines in PLH, including feasibility, prospects, limitation and current development

Response: Thank you for the suggestion. We have added Table 4 that describes possible strategies to improve vaccine response in PLH, including potential benefits and limitations as well as feasibility.

Round 2

Reviewer 1 Report

The revision is well done and acceptable for publication in its current form. I recommend acceptance for publication.

Reviewer 2 Report

The manuscript has been significantly improved for publication.

No.